# Biophysical properties of single rotavirus particles account for the functions of protein shells in a multilayered virus

Manuel Jiménez-Zaragoza[1], Marina PL Yubero[1], Esther Martín-Forero[2], Jose R Castón[3], David Reguera[4], Daniel Luque[2]*, Pedro J de Pablo[1,5]*, Javier M Rodríguez[2]*

[1]Departamento de Física de la Materia Condensada, Universidad Autónoma de Madrid, Madrid, Spain; [2]Centro Nacional de Microbiología/ISCIII, Majadahonda, Spain; [3]Department of Structure of Macromolecules, Centro Nacional de Biotecnología/CSIC, Madrid, Spain; [4]Departament de Física de la Matèria Condensada, Facultat de Física, Universitat de Barcelona, Barcelona, Spain; [5]Instituto de Física de la Materia Condensada (IFIMAC), Universidad Autónoma de Madrid, Madrid, Spain

**Abstract** The functions performed by the concentric shells of multilayered dsRNA viruses require specific protein interactions that can be directly explored through their mechanical properties. We studied the stiffness, breaking force, critical strain and mechanical fatigue of individual Triple, Double and Single layered rotavirus (RV) particles. Our results, in combination with Finite Element simulations, demonstrate that the mechanics of the external layer provides the resistance needed to counteract the stringent conditions of extracellular media. Our experiments, in combination with electrostatic analyses, reveal a strong interaction between the two outer layers and how it is suppressed by the removal of calcium ions, a key step for transcription initiation. The intermediate layer presents weak hydrophobic interactions with the inner layer that allow the assembly and favor the conformational dynamics needed for transcription. Our work shows how the biophysical properties of the three shells are finely tuned to produce an infective RV virion.
DOI: https://doi.org/10.7554/eLife.37295.001

*For correspondence:
dluque@isciii.es (DL);
p.j.depablo@uam.es (PJP);
j.rodriguez@isciii.es (JMR)

**Competing interests:** The authors declare that no competing interests exist.

## Introduction

The advent of single-molecule techniques have opened the door to understand how the mechanics of biomolecular assemblies is essential for their function (*Howard, 2001*; *Müller et al., 2002*). In the case of viruses, the infectious particle must be robust enough to protect the viral genome outside the cell but also competent to undergo the required structural changes once the host cell is recognized, overcome its barriers and carry out the events necessary for a productive viral replication cycle (*Flint et al., 2004*).

Double-stranded RNA (dsRNA) viruses have a number of common challenges derived from the very nature of their genome. Specifically, since there are no host cell enzymes that can recognize dsRNA as template for transcription, the viral particle must incorporate a transcription machinery able to synthesize the required mRNAs to initiate the viral replication cycle. In addition, dsRNA is an inducer of the innate cell-based antiviral response, including interferon synthesis and apoptosis (*Mertens, 2004*; *Arnold et al., 2013*). The virus must evade the host sentinels that trigger these mechanisms and control the host response (*Akira et al., 2006*; *Frias et al., 2012*). Most dsRNA viruses exhibit a common solution to these problems, which consists of the assembly of a stable protein cage in the host cytoplasm that isolates the viral dsRNA molecules to prevent the cellular

**eLife digest** Viruses are small agents that enter and hijack cells to create more of themselves. Most of them are made of a protein shell that encases the viral genome and certain molecular tools. During the life cycle of a virus, this shell fulfils many roles, from protecting the genetic information to recognising the appropriate host cell. It must also disassemble at the right time for replication to take place.

A number of viruses wrap themselves in several layers of protective casing, resulting in an onion-like structure. For example, the rotaviruses that sometimes cause severe diarrhoea in young children have three layers, each with specific properties. Rotavirus subparticles may exist with only one or two of these coats, which allows researchers to study each layer in detail.

Here, Jiménez-Zaragoza et al. use a method called atomic force microscopy to look into the physical properties of the layers of the rotavirus. The technique uses an extremely sharp stylus attached to a tiny cantilever to deform the shells of a single virus. How the structure reacts can then be recorded using a powerful microscope. This helps to determine the stiffness of the layers, and how much force is required to break or weaken each of them.

The experiments reveal that the mechanical properties of the layers are tailored to help the virus survive and infect cells. The outer coat is stiff and resistant to strain, which shields the virus during its travel through the digestive system. The middle layer is the thickest and the softest of the three. It allows the virus to cope with deformation, which is necessary for the expression of its genome.

The outer and middle layers are strongly connected, in part through calcium ions that may be 'sandwiched' between the two. By contrast, the middle and inner layers are only loosely attached to each other. When the virus enters the cell, the calcium ions get dislodged, helping the external coating to easily disassemble. In turn, this creates structural changes in the middle layer, which activate molecules required for the genome to get expressed. Ultimately, disrupting the finely tuned properties of the layers could create new ways of fighting rotaviruses.

DOI: https://doi.org/10.7554/eLife.37295.002

antiviral response. This cage (the viral core) incorporates the necessary enzymes for transcription and replication of the dsRNA genome, which are accomplished without disassembly the particle. This core presents a common architecture that consists of an icosahedral T = 1 shell formed by 60 asymmetric dimers (a 120-subunit capsid) (*Jaing et al., 2008*) present in most of the dsRNA virus families (*King et al., 2011*). Most of these viruses present a single protein shell and lack an extracellular cycle (*Ghabrial et al., 2015*). However, *Cystoviridae* and *Reoviridae* families display concentric protein layers surrounding the core that are responsible of host cell recognition, entry, etc. This modularity facilitates the study of the relationship between the layer functions, their structure and physical properties.

RV, a major causative agent of severe dehydrating diarrhea in children under five years (*GBD Diarrhoeal Diseases Collaborators, 2017*), is a multilayered virus of clinical relevance and one of the main study models for the *Reoviridae* family. The RV infectious particle is a 100 nm non-enveloped triple-layered particle (TLP) composed of three concentric protein shells enclosing the dsRNA genome and the viral RNA polymerase and capping enzyme (*Figure 1A*) (*Settembre et al., 2011*). The inner layer is a T = 1 capsid formed by 60 asymmetric dimers of the VP2 protein (102 kDa) that surrounds the eleven dsRNA genomic segments associated with the RNA-dependent RNA-polymerase VP1 (125 kDa) and the RNA-capping enzyme VP3 (88 kDa) at the pentameric positions (*Estrozi et al., 2013*; *Periz et al., 2013*). This thin single-layered particle (SLP), an intermediate structure that is involved in the packing and replication of the viral genome, is surrounded by a thick T = 13 layer formed by 260 VP6 pear-shaped trimers (45 kDa) (*Settembre et al., 2011*; *McClain et al., 2010*) in the double-layered particle (DLP). This particle, which does not disassemble during the infection, constitutes the transcriptional machinery that initiates the core steps of the viral replication cycle once delivered in the host cell cytoplasm (*Cohen et al., 1979*; *Bass et al., 1992*; *Lawton et al., 1997*). The DLP is not infectious since it cannot recognize, bind to and penetrate the host target cell. These abilities are incorporated in the outer layer of the TLP formed by VP4 and VP7. The VP7 glycoprotein is organized as 260 $Ca^{2+}$-stabilized trimers that cap and embrace through

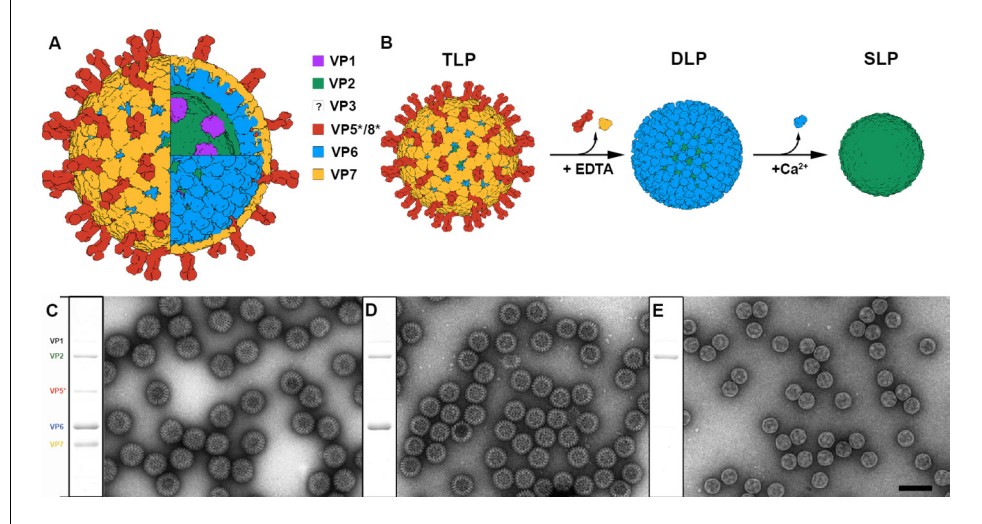

**Figure 1.** Production and purification of TLP and subviral particles. (**A**) Schematic representation of the mature RV TLP. Color code is detailed. (**B**) DLP and SLP generation from TLP. VP7 and VP5*/VP8* are disassembled from TLP in the presence of EDTA. High concentration of $Ca^{2+}$ ions takes apart VP6 trimers to liberate SLP. (**C–E**) Coomassie blue-stained SDS-PAGE gels and negative staining electron microscopy of TLP (**C**), DLP (**D**) and SLP (**E**). Positions of rotavirus structural proteins (VP) are indicated. The question mark indicates the unknown position and structure of VP3. The bar represents 100 nm.

DOI: https://doi.org/10.7554/eLife.37295.003

The following figure supplement is available for figure 1:

**Figure supplement 1.** Cryo-EM analysis of TLP.

DOI: https://doi.org/10.7554/eLife.37295.004

its N-terminal arm each VP6 trimer of the DLP (*Settembre et al., 2011*; *Chen et al., 2009*). Sixty spikes are anchored on the VP6 layer depressions that surround the pentameric positions and are clamped by the VP7 layer. The viral spike is formed by three copies of VP4 that must be proteolytically processed to VP5* and VP8* by trypsin-like proteases from the intestinal lumen or from within cells to generate a fully-infectious virion (*Settembre et al., 2011*; *Estes et al., 1979*; *Estes et al., 1981*; *Clark et al., 1981*). Interestingly, the assembly of the VP6 T = 13 layer on the 60 VP2 dimers (T = 1) that build the SLP is one of the best examples of symmetry mismatch, of which the consequences for virus particle stability are still not well understood. This mismatch is preserved in most reoviruses, and has been associated with the regulation of the polymerase activity (*McClain et al., 2010*). In contrast with the plethora of information obtained during 30 years of structural studies on the particle components (*Trask et al., 2012*), little is known about the mechanical properties of the RV particle layers, subviral particles and TLP, and their mutual influence in contributing to the virus stability along its cycle. Both the application (*Rief et al., 1997*; *Perrino and Garcia, 2016*) and measurement (*Hua et al., 2002*; *Alsteens et al., 2017*) of forces on single molecules are key methodologies to decipher the function of biomolecular systems. Specifically, the study of viral capsids by Atomic Force Microscopy (AFM) enables the exploration of physicochemical properties, such as mechanics and electrostatics, in liquid milieu by using a sharp tip attached to a cantilever to probe individual particles (*Roos et al., 2010*). Single indentation assay consists on deforming a virus particle with the AFM tip while recording the cantilever bending *vs.* the virus deformation to induce the virus breakage (*Roos, 2018*). The force-indentation curves (FIC) so obtained inform about the virus stiffness or spring constant (elasticity), breaking force and brittleness. AFM also allows applying repetitive loading cycles to individual viruses at low force (~100 pN) which results in mechanical fatigue experiments (*Moreno-Madrid et al., 2017*). AFM directly probed the existence of pressure (*Kindt et al., 2001*; *Smith et al., 2001*) in some phages (*Evilevitch et al., 2011*; *Hernando-Pérez et al., 2012*) that is used to translocate their genome into the host (*González-Huici et al., 2004*). In a similar way, it has been found that human adenovirus pressurizes during maturation, and that this pressure is related to the degree of condensation of the dsDNA of the viral

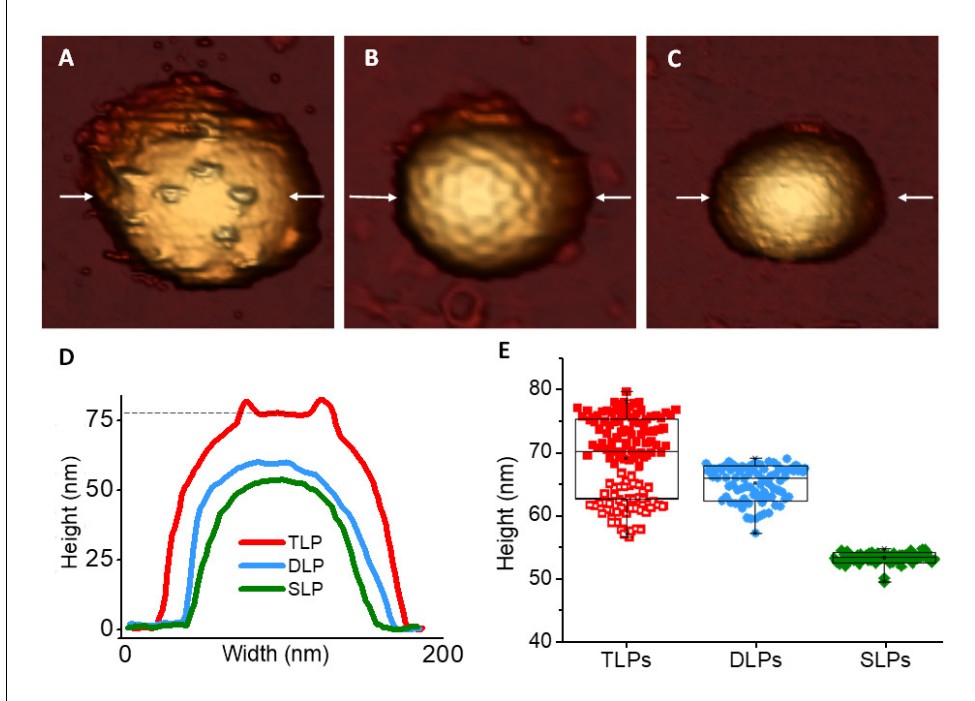

**Figure 2.** AFM topographies of TLP and subviral particles. (**A–C**) AFM images of TLP (**A**), DLP (**B**) and SLP (**C**). (**D**) Height profile of the TLP, DLP and SLP taken along the central part (indicated with arrows) of the particles shown in panels A-C. Dashed line indicates the height of the particle obtained at the VP7 layer. (**E**) Box plot of heights measured from single TLP [69.7 ± 6.1 nm (red, N = 129)], DLP [65.7 ± 2.8 nm (blue, N = 82)] and SLP [53.8 ± 0.9 nm (green, N = 71)]. The two different populations of TLP are indicated with filled and empty red squares (see main text). Height data are available from *Figure 2—source data 1* and *2*.

DOI: https://doi.org/10.7554/eLife.37295.005

The following source data and figure supplements are available for figure 2:

**Source data 1.** Topo profiles of *Figure 2D*.
DOI: https://doi.org/10.7554/eLife.37295.008
**Source data 2.** Height data points statistics of *Figure 2E*.
DOI: https://doi.org/10.7554/eLife.37295.009
**Figure supplement 1.** AFM topography of TLP.
DOI: https://doi.org/10.7554/eLife.37295.006
**Figure supplement 2.** Deformation of RV subviral particles after adsorption on HOPG.
DOI: https://doi.org/10.7554/eLife.37295.007

minichromosome (*Ortega-Esteban et al., 2015a*; *Ortega-Esteban et al., 2015b*). In addition, the influence of both homologous (*Mertens et al., 2015*; *Zeng et al., 2017a*) and heterologous (*Llauró et al., 2016a*; *Snijder et al., 2016*) cargos have been explored in virus mechanics. The alteration of the capsid structure with maturation (*Roos et al., 2012*; *Hernando-Pérez et al., 2014a*), mutations (*Castellanos et al., 2012*; *van Rosmalen et al., 2018*) or cementing proteins (*Hernando-Pérez et al., 2014b*; *Llauró et al., 2016b*) also influences virus mechanics. However, these studies have been never applied to multilayered virus particles, which enable direct measurements of the inter-layer interactions magnitude. Here, we explore the mechanical properties of individual TLP, DLP and SLP particles by single indentation assay and probe their stability against mechanical fatigue. Our experiments, in combination with Finite Element (FE) analysis, the atomic structure of the layers and the calculation of the electrostatic properties of each particle, allow to probe and interpret the intra and interlayer interactions and relate them to their role during the virus replication cycle.

# Results

## Purification and characterization of TLP, DLP and SLP

Previous studies have shown that RV TLP can be converted to DLP by disassembling the outer VP4-VP7 layer with chelating agents such as ethylenediaminetetraacetic acid (EDTA) (*Estes et al., 1979*). Once purified, DLP can be converted to SLP by chaotropic agents such as $CaCl_2$ (*Figure 1B*) (*Bican et al., 1982*). TLP were purified from infected cells, and DLP and SLP were produced and purified combining the above described treatments with several ultracentrifugation steps to remove the proteins of the disassembled layers. Homogeneous populations of TLP, DLP and SLP were obtained, as indicated by SDS-PAGE and negative staining electron microscopy analysis (*Figure 1C–E*). Spike polypeptides (VP5*/VP8*) and VP7 glycoprotein are totally removed in purified DLP (*Figure 1D*) while VP6 is absent in the isolated cores (*Figure 1E*).

## AFM topography of TLP and RV subviral particles

After the adsorption of particles on substrate, we used AFM in *jumping mode* (*Ortega-Esteban et al., 2012*) for the topographical characterization of individual particles in liquid. Our high resolution images (*Figure 2*) are compatible with the structures obtained from cryo-EM (*Settembre et al., 2011*; *Zhang et al., 2008*) and x-ray (*McClain et al., 2010*), where thousands and millions of particles are averaged, respectively. Spikes protruding from the TLP (*Figure 2A*) as well as the DLP pentameric and hexameric depressions (*Figure 2B*) are resolved. In contrast, SLP offers featureless structure (*Figure 2C*). Although the distinctive topography of TLP allows their unambiguous identification, they exhibit a broad distribution of height values (*Figure 2D–E*). This behavior is probably due to the number of spikes (*Figure 2—figure supplement 1*) present at the interface between the particles and the substrate surface and to the mode of how they influence the particle adsorption. The height data of TLP (*Figure 2E*, red) suggest two populations centered at ~74 and ~62 nm, represented by filled and empty symbols, respectively. We propose that these data correspond to the presence (red filled squares, *Figure 2E*) or absence (red empty squares, *Figure 2E*) of spikes at the particle-surface interface. In the first case the presence of spikes would prevent partially the contact between the VP7 layer and the substrate (*Figure 2—figure supplement 2*), thus precluding virus adsorption and deformation. However, when the VP7 layer directly rests on the surface, TLP collapse to an average height value of 62 nm (*Figure 2*) probably due to a strong VP7-surface interaction (*Zeng et al., 2017a*). In contrast, DLP and SLP present a narrower height distribution (*Figure 2E*) whose average values are compatible with the nominal values (70 nm for DLP and 55 nm for SLP), indicating a little deformation due to the adsorption on the surface of 6% and 2% for DLP and SLP respectively (*Llauró et al., 2015*; *Zeng et al., 2017b*).

Although icosahedral symmetry imposition renders an ideal RV particle with 60 trimeric spikes (*Figure 1A* and *Figure 1—figure supplement 1*), previous studies have shown that some positions are unoccupied in the purified TLP (*Chen and Ramig, 1992*; *Trask and Dormitzer, 2006*; *Rodríguez et al., 2014*). To estimate the amount of spike protein in TLP, VP5* was quantified relative to protein VP6 (occupancy). Densitometric analysis of Coomassie-stained gels (*Figure 1C*) produced an occupancy of 52%. Cryo-EM analysis and three-dimensional reconstruction (3DR) of these TLP showed an equivalent occupancy (~50%) when the relative density of the spikes in the 3DR is determined using the VP2-VP6-VP7 shell density as a reference (*Figure 1—figure supplement 1*). This occupancy correlates with the different number of spikes detected in the AFM images of single TLP (*Figure 2—figure supplement 1*). Since lateral spikes are easily removed by the AFM tip (*Video 1*), we analyzed the upper ~1/3 region of the virus surface, where the spikes point upwards and present a maximum resistance to AFM imaging. Although we cannot ignore that the AFM tip could remove some

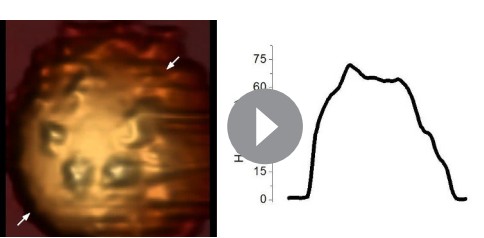

**Video 1.** Mechanical fatigue over TLP in TNC buffer during 32 frames (~82 min) at 100–200 pN per pixel (1 pixel = 1.4–2.3 nm). This video corresponds to the particle of *Figure 5A*. White arrows indicate the line where the profiles have been obtained.
DOI: https://doi.org/10.7554/eLife.37295.010

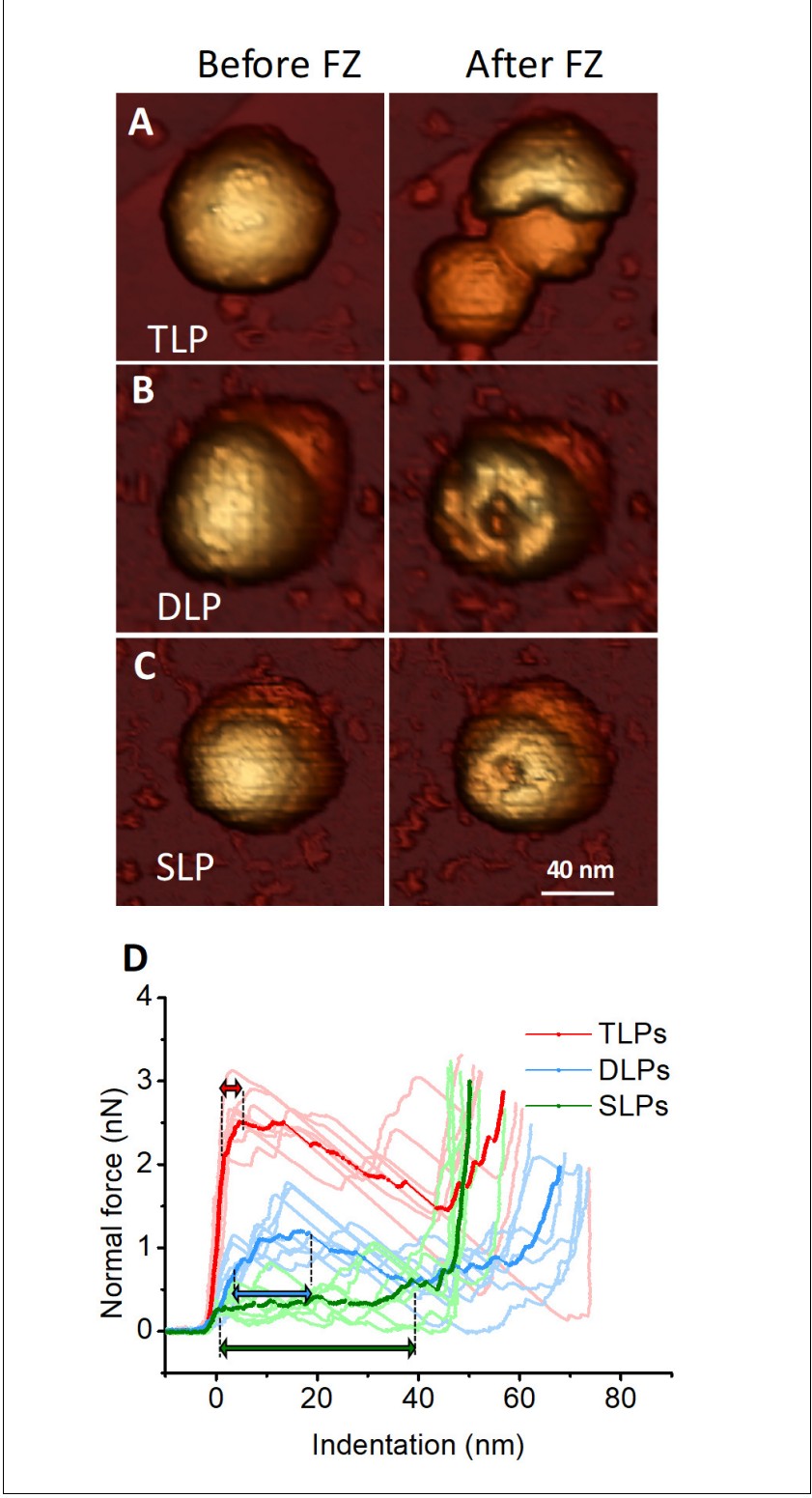

**Figure 3.** Single-indentation assay of TLP and subviral particles. (A–C) AFM topographies of an individual TLP (**A**), DLP (**B**) and SLP (**C**) before (left) and after (right) nanoindentation. (**D**) Force Indentation Curves (FICs) measured for each individual TLP (N = 7 from 45), DLP (N = 7 from 11) and SLP (N = 7 from 16), as indicated. The average curve is highlighted for each specimen. Double headed arrows indicate plastic deformation, as explained in text. FIC data are available from *Figure 3—source data 1*.

DOI: https://doi.org/10.7554/eLife.37295.011

*Figure 3 continued on next page*

*Figure 3 continued*

The following source data is available for figure 3:

**Source data 1.** Indentation curves.

DOI: https://doi.org/10.7554/eLife.37295.012

spikes, we minimized this effect by using the first image obtained for each particle. Our AFM topographies, which uniquely allow for the first time the direct imaging of the individual spikes, provide a more realistic view of the RV virion as a distribution ranging from fully decorated to almost naked particles. (*Figure 2* and *Figure 2—figure supplement 1*). We can directly observe an average occupancy of 35%, compatible with electrophoretic and cryo-EM bulk analysis results. These data support the in vitro recoating assays, demonstrating that an occupancy as low as 10% is enough to generate particles with high specific infectivity (*Trask and Dormitzer, 2006*).

## Single indentation assay

In order to investigate the contribution of the different layers to the mechanical stability of the RV particle, systematic single indentation experiments of the different particles were performed (*Figure 3*) resulting in broken structures. In order to understand the nature of each particle breakage it is interesting to compare their topographies before and after fracture (*Figure 3A–C*), and to consider the average indentation curves for each type of structure (*Figure 3D*, strong colors). While TLP breaks into large fragments (*Figure 3A*, right), both DLP and SLP show circular deformations that can be attributed to the tip apex (*Figure 3B–C*, right). The average of TLP nanoindentation curves (*Figure 3D*, strong red) shows a linear regime that corresponds to the virus elastic deformation up to ~2.0 nm at ~2.1 nN, where the elastic limit is reached. Afterwards the structure yields plastically without breaking until 2.5 nN at 4.7 nm, during 2.7 nm (red double headed arrow, *Figure 3D*), where the downwards slope indicates fracture. The same reasoning applied to both DLP and SLP result in plastic deformations of ~13 nm and ~39 nm, respectively (blue and green double headed arrows, *Figure 3D*). Virus topographies and indentation assays indicate that while TLP undergoes a brittle (glass-like) fracture, both DLP and SLP experience ductile (rubber-like) breakage.

## Stiffness and yield strain of TLP and RV subviral particles

The analysis of single particle FIC charts (*Figure 3D*) provides some mechanical parameters. In particular, the linear fitting of the curves before reaching the elastic limit informs about the particle elastic constant or stiffness (*Figure 4A*). Statistical analysis of the FIC linear part result in spring constants of $k_{TLP} = 0.76 \pm 0.30$ N/m, $k_{DLP} = 0.34 \pm 0.20$ N/m and $k_{SLP} = 0.22 \pm 0.07$ N/m. The elastic limit can be linked to the breaking force of the probed particle. The analysis of the breaking force provides values of $2.9 \pm 0.5$ nN, $0.9 \pm 0.3$ nN and $0.45 \pm 0.10$ nN for TLP, DLP and SLP, respectively (*Figure 4B*). This monotonic decrease of both the spring constant and breaking force with the reduction of the number of layers indicates that virus mechanics captures the reinforcement nature of concentric shells: the more layers in the structure, the stronger it becomes. The calculation of the yield strain $\varepsilon = \frac{\Delta h}{h_0}$ (*Figure 4—figure supplement 1*), where $\Delta h$ is the indentation corresponding to the force at the elastic limit and $h_0$ the height of the intact particle, reveals that TLP, although with a high dispersion, can sustain larger elastic deformations than DLP and SLP. This simultaneous high rigidity and yield strain is exceptional since an increment in the spring constant and breaking force is usually associated with a lower yield strain as it happens, for example, with glass (*Schijve, 2009*).

## Finite elements analysis of TLP and RV subviral particles

The analysis of how the TLP and DLP inform upon the mechanical properties of VP6 and VP7 layers has to be considered with care. The only layer for which an individualized analysis of the mechanical properties can be performed is the SLP. Although the genome and the replication/transcription machinery reside inside the VP2 shell, it is expected that they have no relevant effect on the particle's response to deformation. The relatively low packing fraction of RV (~20%) compared to pressurized dsDNA viruses (*Purohit et al., 2005*) suggests a small pressure whose influence on the effective elastic constant will be smaller than our error bars. In any case, the presence of the core would only affect to the estimation of the Young's Modulus of the VP2 layer, but not to the inferred

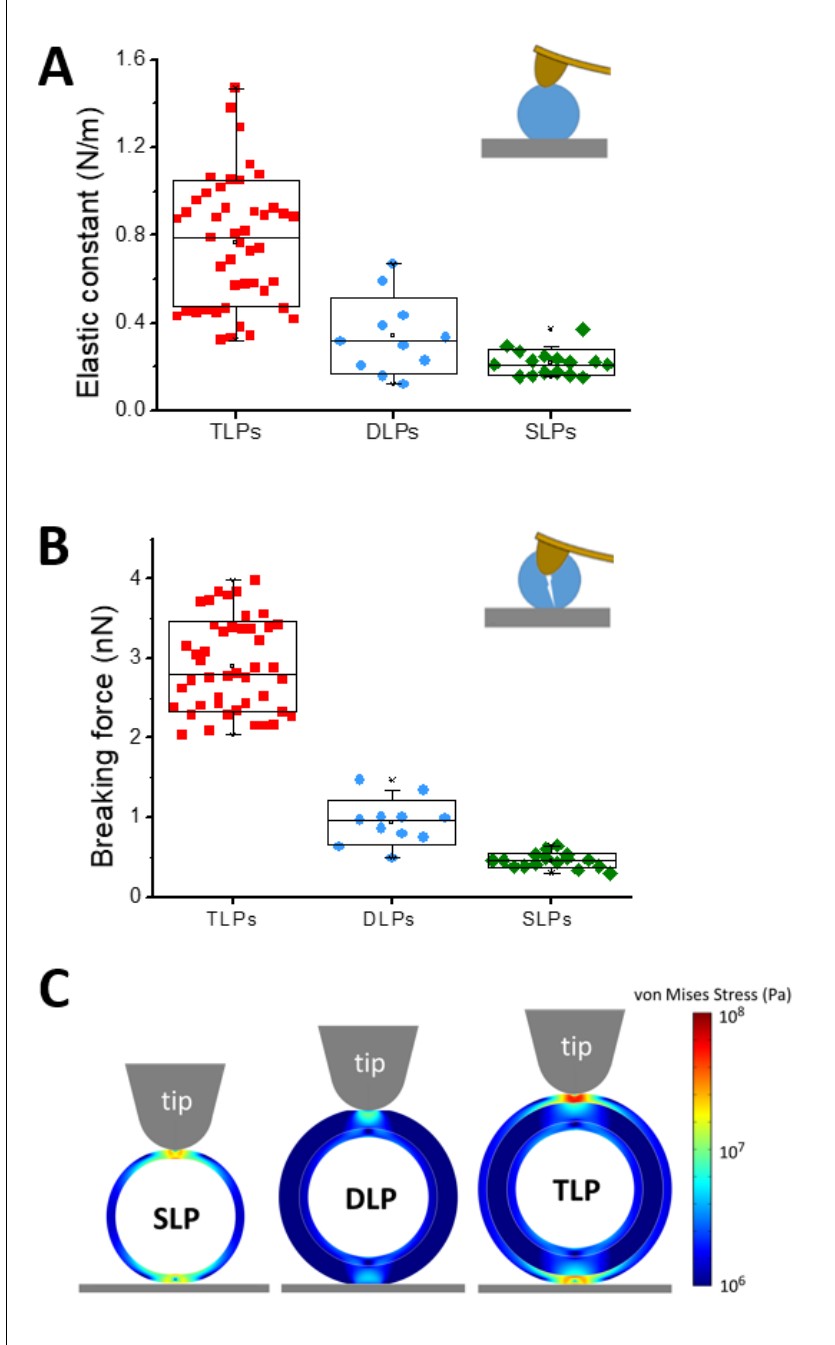

**Figure 4.** Mechanical properties. Box plots of (**A**) elastic constant, and (**B**) breaking force calculated from the FIC (*Figure 3*) for TLP (red, N = 45), DLP (blue, N = 11) and SLP (green, N = 16), as explained in text. Data are *Figure 4—source data 1* and *2*. The yield strain ($\varepsilon$) which is a combination of elasticity and breaking force, can be found in *Figure 4—figure supplement 2*. (**C**) Stress distribution at 4 nm of indentation in the models used in the FE simulations of SLP (left), DLP (middle) and TLP (right). Inset color scale represents the value of the von Mises stress in Pa. See Materials and methods and *Figure 4—figure supplement 1* for more information.

DOI: https://doi.org/10.7554/eLife.37295.013

The following source data and figure supplements are available for figure 4:

**Source data 1.** Spring constants of *Figure 4A*.
DOI: https://doi.org/10.7554/eLife.37295.017
**Source data 2.** Breaking forces of *Figure 4B*.
DOI: https://doi.org/10.7554/eLife.37295.018
*Figure 4 continued on next page*

*Figure 4 continued*

**Figure supplement 1.** Finite element model of the RV particle.
DOI: https://doi.org/10.7554/eLife.37295.014
**Figure supplement 2.** Yield strain.
DOI: https://doi.org/10.7554/eLife.37295.015
**Figure supplement 2—source data 1.** Critical strain of *Figure 4—figure supplement 2*.
DOI: https://doi.org/10.7554/eLife.37295.016

properties of the VP6 and VP7 layers. For TLP and DLP, the isolation of the mechanical parameters for VP7 and VP6 layers is also complex because they include internal shells with their mutual interactions. Specifically, the mechanical response of DLP is due to the VP6 shell and the internal SLP, whereas in the TLP there is an additional contribution of the VP7 layer. Taking this into account, FE simulations (*Gibbons and Klug, 2008*) were performed to extract the effective Young's moduli for the different capsid layers from the measured spring constants $k_{SLP}$, $k_{DLP}$, $k_{TLP}$ (see Materials and methods and *Figure 4—figure supplement 1*). The nanoindentation of SLP was implemented first, yielding a value for the Young's modulus of $Y_{VP2}$ = 0.53 ± 0.20 GPa. A second layer of 8 nm thickness, representing that of VP6, was added on top of the VP2 layer, and a Young's modulus $Y_{VP6}$ = 0.08 ± 0.07 GPa was needed to recover the spring constant of DLP, $k_{DLP}$. Finally, a 3.5 nm thick third layer was placed on top of the DLP, requiring a Young's modulus $Y_{VP7}$ = 1.0 ± 0.9 GPa to yield the same spring constant as the TLP, $k_{TLP}$. We can compare the Young's modulus between layers resulting in $Y_{VP7}$ / $Y_{VP6}$=12 and $Y_{VP2}$ / $Y_{VP6}$=6.5. *Figure 4C* shows the map of the stress supported by the constituent layers of each subviral particle, demonstrating that the VP7 layer accumulates most of the stress in TLP. Thus, nanoindentation experiments and FE analysis, indicate the VP7 shell to be the stiffer layer of the RV structure, but also the most elastic, whereas the thick VP6 layer is remarkably soft and brittle.

## Mechanical fatigue

While the single indentation assay probes the global mechanical response of virus particles, fatigue experiments explore the local response of the virus building blocks (capsomers). Mechanical fatigue experiments are performed by applying cyclic forces of ~100 pN, well below the breaking force (~1 nN), at every pixel of the virus (*Ortega-Esteban et al., 2013*) and the gradual disassembly of viral particles is typically induced (*Hernando-Pérez et al., 2014a*). Cyclic imaging of the TLP (*Figure 5A*, left) at forces between 100 pN to 200 pN per pixel shows that, while the VP4 spikes are removed from the particle surface in a few frames (*Figure 5A*, middle and *Video 1*), the VP7 layer remains mostly intact (*Figure 5A*, right) during 80 frames (light red in *Figure 5E*, *Figure 5—figure supplement 1* and Video 5). These results illustrate that the spikes are easily removed by the AFM tip and are not strongly anchored. However, the VP7 layer displays a strong resistance against fatigue, in agreement with the high stiffness and breaking force demonstrated in single indentation assay experiments. A strong binding energy between capsomers would not only result in a high resistance of individual proteins against fatigue, but also will contribute to a high breaking force when all capsomers are probed in a single indentation assay experiment. We have found similar results before in lambda phage (*Hernando-Pérez et al., 2014a*).

The current model proposes a calcium concentration drop in endosomal compartments during RV entry as the factor that triggers VP7 disassembly and membrane penetration (*Arias et al., 2015*). In fact, calcium depletion by chelating agents (as EDTA) (*Estes et al., 1979*) is used to uncoat TLP to DLP by inducing VP7 trimer dissociation (*Figure 1*). To explore the structural consequences of this process in real time, we carried out fatigue assays on TLP while EDTA simultaneously flowed in the AFM liquid chamber, as described in Material sand methods, to induce the gradual depletion of Ca ions of the particles (*Figure 5B*, *Video 2*). In these conditions, fatigue induces the neat VP7 detachment from the VP6 subjacent layer (indicated by a circle in *Figure 5B*#19) even before the spikes are removed. Indeed, the evolution of the topographic profiles (dark red in *Figure 5E*) show abrupt downwards steps very close to the VP7 thickness (red arrow of *Figure 5E*) indicating that TLP particle loses VP7 completely while keeping VP6 (*Video 2*). These results not only suggest that Ca ions mediate the interaction between VP7 and VP6 layers, but also that the absence of ions weakens the interaction between VP7 subunits. If fatigue continues, VP6 subunits are neatly removed from VP2

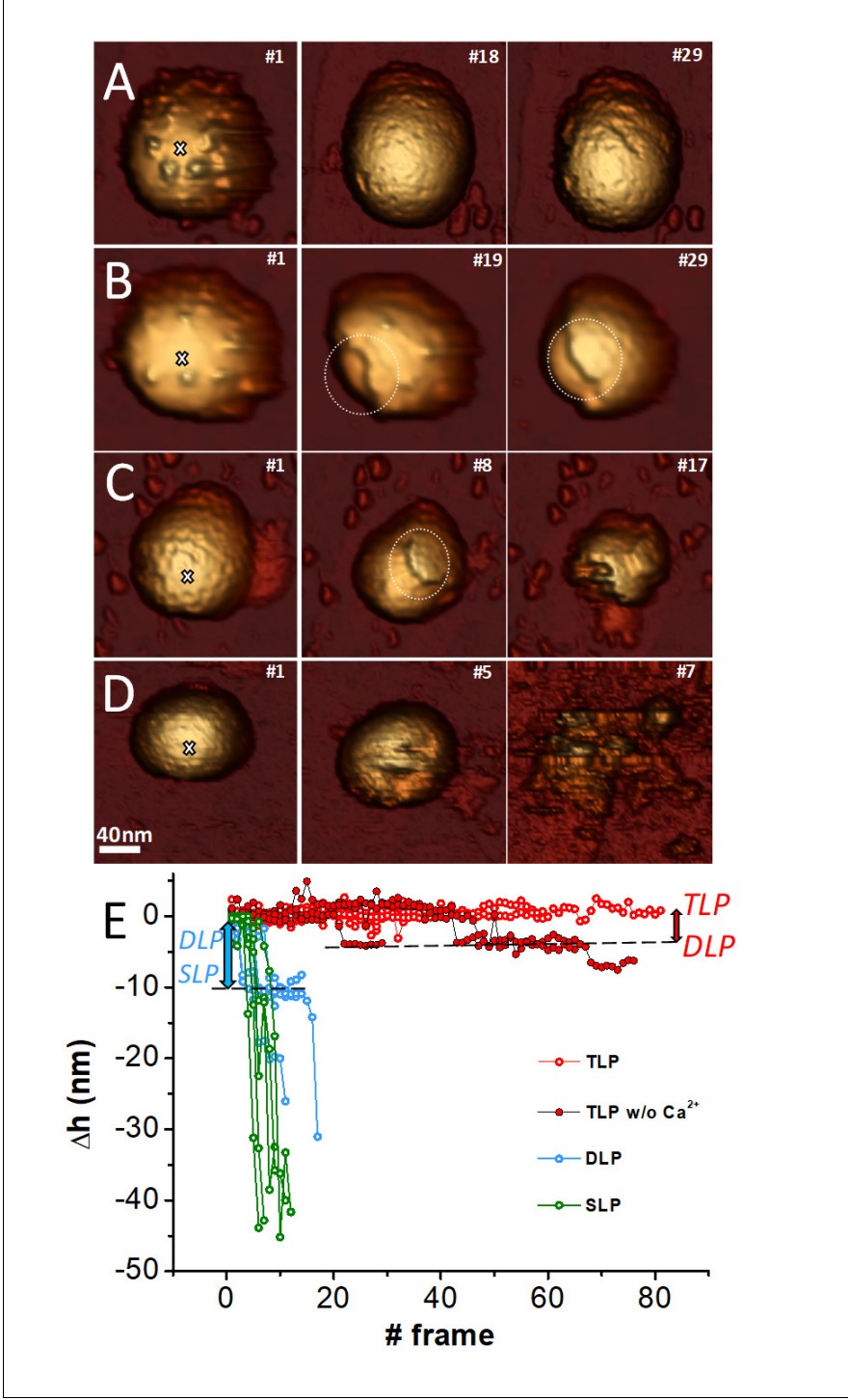

**Figure 5.** Fatigue of TLP and subviral particles. Topographic evolution of TLP (**A**), TLP + EDTA (**B**), DLP (**C**) and SLP (**D**) during continuous imaging at low force (~60–120 pN) indicating the corresponding displayed frames. (**E**) Topographic evolutions obtained at the position indicated with a white cross (**A–D**) in TLP (red), DLP (blue) and SLP (green) particles. Dark red color indicates fatigue of TLP + EDTA. Red and blue arrows indicate the loss of height from TLP to DLP and for DLP to SLP, respectively. Data are available from *Figure 5—source data 1*. *Videos 1–4* present the temporal evolution of these particles.

DOI: https://doi.org/10.7554/eLife.37295.019

The following source data and figure supplements are available for figure 5:

**Source data 1.** Data height evolution with mechanical fatigue.

*Figure 5 continued on next page*

*Figure 5 continued*

DOI: https://doi.org/10.7554/eLife.37295.022

**Figure supplement 1.** Fatigue of TLP.

DOI: https://doi.org/10.7554/eLife.37295.020

**Figure supplement 1—source data 1.** Profiles of the *Figure 5—figure supplement 1B*

DOI: https://doi.org/10.7554/eLife.37295.021

---

layer (circle in *Figure 5B*#29). Therefore, VP6 shell appears as a weak shell whose interaction with the beneath VP2 layer is not very strong, since it peels off rapidly to reveal the SLP.

We found similar results on experiments performed on DLP. Again, fatigue induced a clean VP6 disassembly after less than 10 frames (circle in *Figure 5C*#8, and *Video 3*). In this case the evolution of the topographic profiles (blue in *Figure 5E*) undergoes sharp reductions very close to the VP6 thickness, inducing the gradual uncovering of the innermost VP2 (blue arrow in *Figure 5E*). These experiments not only illustrate a weak interaction between VP6 and VP2 layers, but also a very feeble VP6-VP6 binding force. Finally, the thin SLP VP2 is highly unstable under fatigue experiments collapsing well before reaching 10 frames (green in *Figure 5D*, and *Video 4*).

## RV TLP nanoindentation-fatigue combined analysis

We have seen that removing of Ca ions is key for inducing the transition from TLP to DLP in the fatigue experiments (*Figure 5A–B*, and red charts in *Figure 5E*). To access to VP6 and VP2 layers in the presence of Ca ions, we combined single indentation with fatigue assays. Our aim is to produce local disruptions in the TLP shell by performing a controlled FIC and then monitor the progressive disassembly induced by fatigue experiments. Therefore, we intent to crack the three layers at once without tearing apart the particle like in *Figure 3A*, by adjusting the indentation up to 40 nm (data not shown) after imaging the TLP 24 times (*Figure 6A#24*). Right after the FIC (*Figure 6A#25*), the induced fracture reaches a maximum depth of ~23 nm that includes the thickness of the three layers (*Figure 6B*). However, the shape of the crack shows that some VP2 layer has been exposed (*Figure 6B*, dotted line) and its distance to the VP7 layer external face is compatible with the thickness of VP7 and VP6 layers (3,5 and 8 nm, respectively). The subsequent fatigue cycles increased the VP2 uncovered area (*Figure 6A#56*, *Video 6*) without any signature of the TLP-DLP transition. These experiments indicate that VP6 hardly survives to VP7 removal, supporting a strong interaction between VP7 and VP6 layers in the presence of Ca and, once again, a weak binding between VP6 and VP2.

## Discussion

The characterization of the biophysical properties of viral particles has proven to be a powerful approach to understand the connection between structure and function in different systems (*Moreno-Madrid et al., 2017*). Our mechanical analysis of the multilayered RV particle offers new opportunities to explore the interplay between structure, function and mechanics. In particular, the atomic structure of the layers provided by X-ray crystallography and cryo-EM (*Settembre et al., 2011*; *McClain et al., 2010*; *Zhang et al., 2008*), allows the discussion of our results at a molecular level. This architecture informs about the interactions among the viral proteins, including the analysis of contact surfaces and their electrostatic nature. Analysis of the electrostatic potential of the different RV particles (*Figure 7A–C*) shows that the core shell presents a mainly hydrophobic outer surface (*Figure 7A*) in agreement with its tendency to

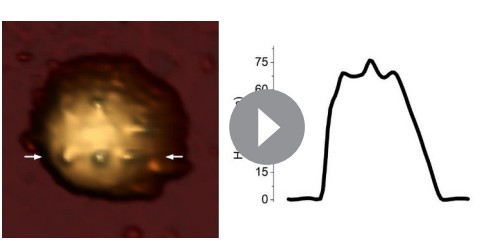

**Video 2.** Mechanical fatigue over TLP in TNC buffer being gradually replaced by TNE buffer (pumping TNE buffer and withdrawing liquid from the sample at 1 μl/min), during 31 frames (~80 min) at 50–60 pN per pixel (1 pixel = 3.1 nm). This video corresponds to the particle of *Figure 5B*. White arrows indicate the line where the profiles have been obtained.

DOI: https://doi.org/10.7554/eLife.37295.023

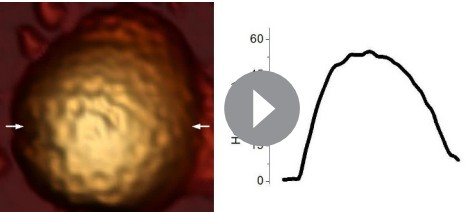

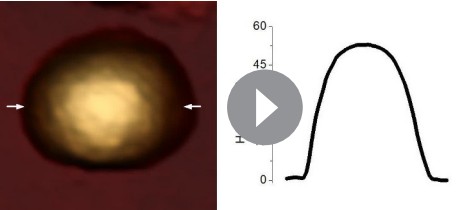

**Video 3.** Mechanical fatigue over DLP in TNC buffer during 17 frames (~45 min) at 60–70 pN per pixel (1 pixel = 2.0 nm). This video corresponds to the particle of *Figure 5C*. White arrows indicate the line where the profiles have been obtained.
DOI: https://doi.org/10.7554/eLife.37295.024

**Video 4.** Mechanical fatigue over SLP in TNC buffer during seven frames (~18 min) at 60–70 pN per pixel (1 pixel = 2.3 nm). This video corresponds to the particle of *Figure 5D*. White arrows indicate the line where the profiles have been obtained.
DOI: https://doi.org/10.7554/eLife.37295.025

form aggregates (*Labbé et al., 1991*; *Desselberger et al., 2013*). While treatment of these cores with electrolytes or different pH do not solubilize them, incubation with some detergents like deoxycholate (*Desselberger et al., 2013*) or with trehalose (*Figure 1E*) disperse them and suggest that particle aggregation is produced by hydrophobic forces. In a RV infection SLP are localized in the viroplasm, where viral RNA packaging and replication occur and where extensive protein-RNA and protein-protein interactions prevent its aggregation (*Zeng et al., 1998*; *Berois et al., 2003*; *Vende et al., 2003*). Over the hydrophobic outer surface of the VP2 T = 1 shell, VP6 pear-shaped trimers assemble into five non-equivalent positions (*Figure 7D–E*, triangles) to build a T = 13 architecture, in what constitutes an extreme example of symmetry mismatch. These mismatched interactions are mainly mediated by the hydrophobic VP6 inward-projecting loop 64–72 (*Figure 7—figure supplement 1*) that contacts with the SLP outer surface, and are not only essential for assembly but also for transcription (*Charpilienne et al., 2002*). Intertrimeric VP6 contacts are established through their pedestal domains and have local 2-fold contacts. Both the VP2-VP6 and the intertrimeric VP6-VP6 contacts are of modest extent. These weak protein-protein interactions, described in the structure, are in agreement with our experiments. In particular, VP6 trimers are quickly disassembled in fatigue experiments (*Figure 5C*; blue in *Figure 5E*), proving poor lateral and perpendicular interactions between VP6 trimers and VP6-VP2 units.

In contrast with the hydrophobic nature of the SLP outer surface, the calculation of the electrostatic potential surface of the DLP reveals a very negative outer surface (*Figure 7B*) (*Mathieu et al., 2001*). The structures of the transcriptionally active particles of other dsRNA viruses present a similar negatively charged outer surface (*Figure 7—figure supplement 2*), which may reflect a common strategy to avoid the interaction with the newly synthesized negative charged transcripts. In addition to transforming the SLP particle in the transcription-active DLP (*Lawton et al., 1997*), the VP6 that polymerizes on the surface of the SLP acts as an adaptor for the interaction with the outer RV shell (*Figure 7F–G*). VP7 trimers are stabilized through the binding of calcium ions at each subunit inter-

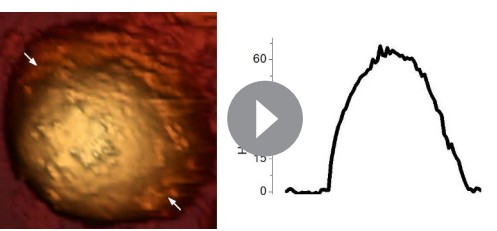

**Video 5.** Mechanical fatigue over TLP in TNC buffer during 81 frames (~180 min) at 60–100 pN per pixel (1 pixel = 2.7 nm). This video corresponds to the particle of *Figure 5—figure supplement 1*.
DOI: https://doi.org/10.7554/eLife.37295.026

face (*Aoki et al., 2009*). The bottom inner surface of the VP7 trimer has minimal contacts with the VP6 trimer apex of which the most intense is mediated by the VP7 N termini that embraces the underlying VP6 trimer (*Figure 7H*). These arms also interact with adjacent VP7 trimers generating a cooperative lattice that reinforce the RV outer shell. Our fatigue experiments (*Figure 5*) demonstrate weak interactions of VP6, both intertrimeric and with the VP2 layer. These analyses also suggest a strong interaction of the VP7 trimers with the underlying VP6 and with the surrounding VP7 trimers in the presence of calcium (*Figure 6*). Many viral particles are stabilized by calcium ions bound to the interfaces between

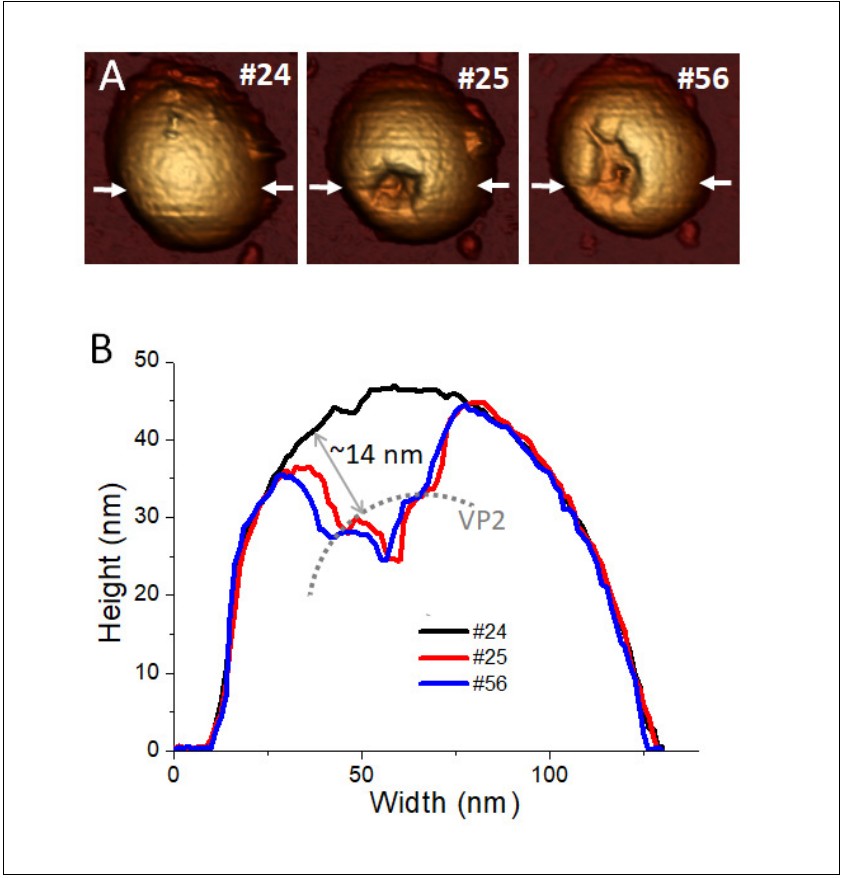

**Figure 6.** Combination of single indentation and fatigue assays. (**A**) Topographical evolution of a TLP subjected to fatigue until image #24, where a controlled FIC locally perforated the three layers. Frame #25 shows the particle right after the FIC. Topography #56 is the same particle 31 frames after the FIC. Imaging force of ~100 pN. (**B**) Height profile evolution obtained at the white arrows depicted in the topographies. Dotted grey line indicates the position of VP2 layer. Data are available from *Figure 6—source data 1*. *Video 6* presents the temporal evolution of this particle.

DOI: https://doi.org/10.7554/eLife.37295.027

The following source data is available for figure 6:

**Source data 1.** Profiles of *Figure 6B*.

DOI: https://doi.org/10.7554/eLife.37295.032

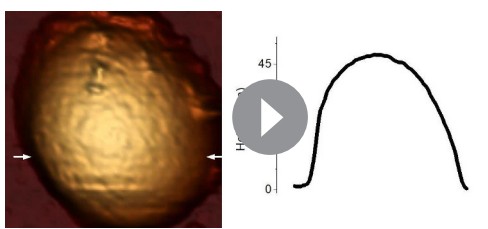

**Video 6.** Mechanical fatigue over TLP in TNC buffer during 72 frames (~155 min) at 100–120 pN per pixel (1 pixel = 1.3 nm). Two moderate nanoindentations of ~ 40 nm were performed at frames 24 and 66, respectively. This video corresponds to the particle of *Figure 6*.

DOI: https://doi.org/10.7554/eLife.37295.028

their capsomers which is allowed by the unique coordination chemistry of the Ca ion (*Zhou et al., 2009*; *Carafoli and Krebs, 2016*). These ions are required to maintain the capsid structural integrity and/or regulate its proper assembly/disassembly (*Zhou et al., 2009*). Examples include bacteriophages of the Leviviridae and Microviridae families (*McKenna et al., 1996*; *Persson et al., 2008*); plant Tombusviruses (and its associate satellite virus), Sobemoviruses, Bromoviruses or Virgaviruses (*Harrison et al., 1978*; *Jones and Liljas, 1984*; *Speir et al., 1995*) and different animal viruses including members of the Polyoma, Noda, Picorna, Birna and Parvoviridae families (*Tsao et al., 1991*). Actually, previous studies have directly probed the mechanical role

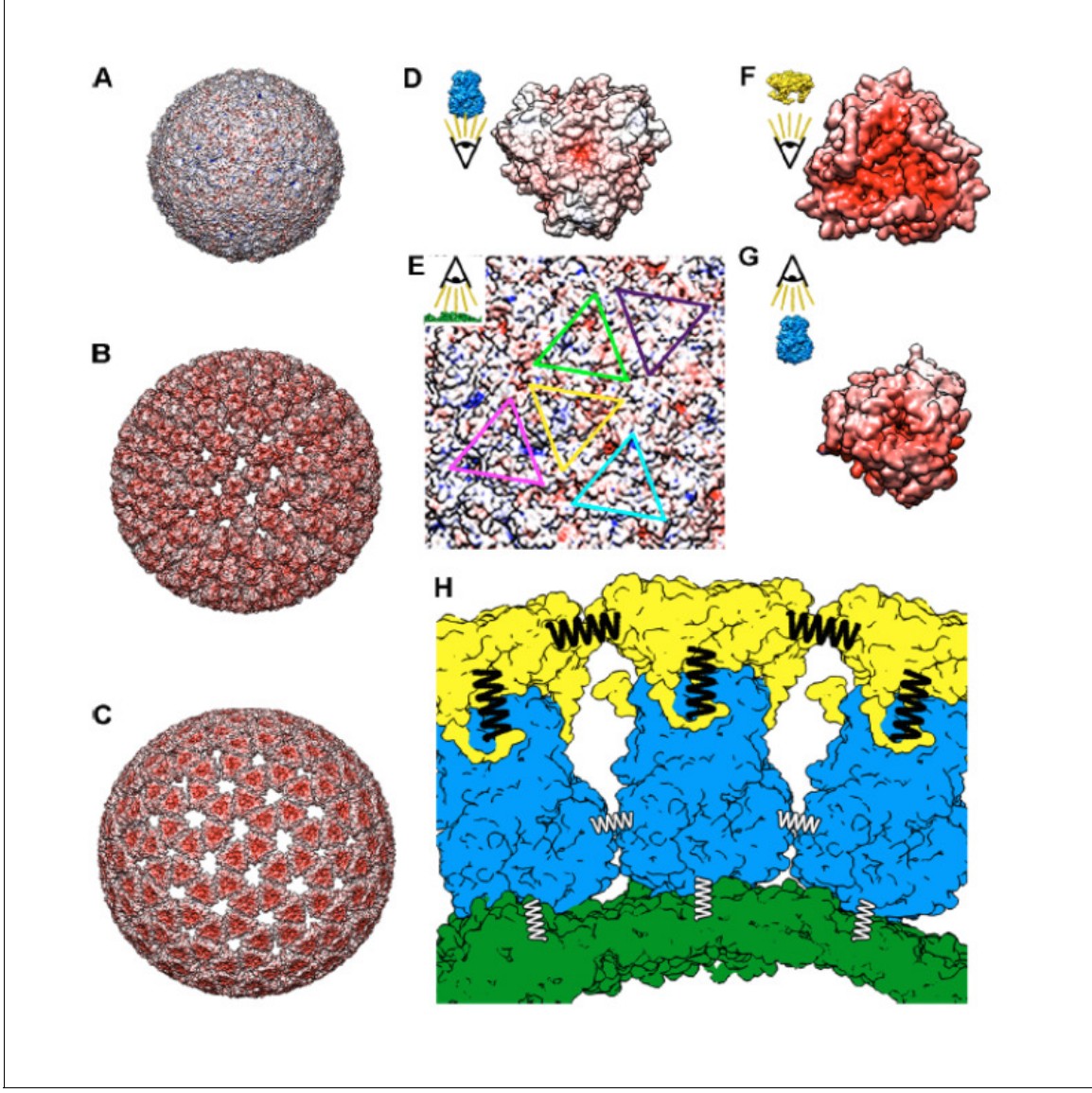

**Figure 7.** Molecular interactions of TLP and subviral particles. Electrostatic potential of the outer surface of SLP (**A**), DLP (**B**) and TLP (**C**). Positive charge distribution is represented in blue, negative in red and hydrophobic in white. (**D**) Electrostatic potential of the inner surface of a VP6 trimer. (**E**) Electrostatic potential of the outer face of the VP2 shell. The positions for the interaction of the five quasi-equivalent trimers on the VP2 surface are marked with triangles of different colors. (**F**) Electrostatic potential of the inner surface of a VP7 trimer. (**G**) Electrostatic potential of the outer surface of a VP6 trimer. (**H**) Schematic representation of the VP2 (green), VP6 (blue) and VP7 (yellow) layers interaction. Thick black springs indicate a relatively high VP7-VP7 and VP6-VP7 interactions. The thin white springs point to weak VP6-VP6 and VP6-VP2 interactions. In panels D-G the insets indicate the point-of-view.

DOI: https://doi.org/10.7554/eLife.37295.029

The following figure supplements are available for figure 7:

**Figure supplement 1.** VP6 trimer atomic structure.
DOI: https://doi.org/10.7554/eLife.37295.030

**Figure supplement 2.** Outer surface of dsRNA virus transcription machineries with electrostatic potential.
DOI: https://doi.org/10.7554/eLife.37295.031

of calcium ions in the shell stability of tomato bushy stunt virus nanoparticles (*Llauró et al., 2015*). Surprisingly, the inward facing electrostatic potential surface of the VP7 layer (*Figure 7F*) is highly negative. We propose that calcium ions, beyond stabilizing the VP7 trimers, would be sandwiched between the VP7 inner and VP6 outer surfaces to allow their assembly. VP7 assembles into trimers

that are stabilized through the binding of two calcium ions at each subunit interface (*Aoki et al., 2009*). Thus, the depletion of calcium will promote the destabilization of the VP7 intertrimeric interactions and induce the rapid disassembly of this shell by the destabilization of the VP7/VP6 electrostatic interactions (*Figure 5B*).

Mechanical parameters, such as stiffness, breaking force and yield strain also inform of important differences between the three layers (*Figure 4*). Similar to that observed for the height distribution (*Figure 2E*), the dispersion detected for TLP stiffness $k_{TLP}$ (*Figure 4A*) could be correlated with the unequal presence and distribution of spikes in each particle. The incorporation of the VP7 layer on the DLP produces a significant increase in stiffness (*Figure 4A*) and yield strain (*Figure 4—figure supplement 2*). Thus, while the Young's modulus value of the VP7 shell is within the highest values as obtained for bacteriophages (*Roos et al., 2012*; *Ivanovska et al., 2004*), the VP6 layer presents the lowest value ever reported for a viral protein shell (*Marchetti et al., 2016*). In fact, the FE simulations of the TLP show that the stiff VP7 layer accumulates most of the stress during the indentation (*Figure 4C*), protecting the internal VP6 and VP2 layers by shielding the stress transmission to these layers. Taken together, nanoindentation and mechanical fatigue experiments demonstrate that the VP7 shell provides the resistance needed by the RV particle to bear with the severe conditions of extracellular media. RV is transmitted through the faecal-oral route and has to overcome the stringent physicochemical conditions of digestion at both the stomach and small intestine, where it infects mature enterocytes (*Estes and Greenberg, 2013*; *Ramig, 2004*). The viscosity of the chyle (*Jonas, 1976*) is about 10 to 100 times larger than the host cytoplasm (*Luby-Phelps et al., 1993*) and presents higher molecular crowding (*Hernando-Pérez et al., 2014a*). Therefore, the VP7 shell has to be stable enough to overcome the constant barrage of molecular impacts in the small intestine. In fact, fatigue experiments provide a good approximation for these molecular impacts on RV particles (*Hernando-Pérez et al., 2014a*). Interestingly, the VP7 shell of TLP is able to withstand fatigue even at 200 pN (*Figure 5A*) indicating a strong intercapsomeric linkage. The labile nature of VP6 layer, showing both the lowest values of elasticity and Young's modulus, is related with their structure (weak contacts of the VP6 trimers with VP2 and between them) and we propose that this feature is necessary for its function. It has been suggested that removal of VP7 causes the dilation of the particle pentameric channels allowing the flux of nucleotides, ions and transcripts (*Chen et al., 2009*; *Aiyegbo et al., 2013*). The removal of VP7 promotes the outward movement of the VP6 pentameric trimers. This conformational change is transmitted through the underlying VP2 decamer to the VP1 polymerase, enabling its activity. In other viruses, such as MVM (*Castellanos et al., 2012*) a similar conformational dynamics is favored by a low mechanical stability. In particular, the increase of local stiffness in MVM mutants blocks the conformational changes required for dsDNA translocation. Similarly, the high flexibility resulting from the low mechanical stability of the trimeric VP6 layer would favor its functional roles: this thick layer becomes the adaptor that allows the transformation of a highly hydrophobic SLP into a negative-charged DLP, overcoming the symmetry mismatch between the T = 1 and T = 13 layers, and generating a transcriptionally active particle. The VP6 shell constitutes the thickest and, according to our data, the softest layer of the RV particle, which allows for large deformations when TLP or DLP are adsorbed (*Figure 2* and *Figure 2—figure supplement 2*). Finally, the SLP exists only in the viroplasm environment during RNA packaging and replication. The high electrodensity of the viroplasm is a signature of a large concentration of macromolecules that results in a higher molecular crowding than the cytoplasm. This fact would explain the higher Young's modulus value of VP2 layer when compared with that of VP6. This Young's modulus combined with a presumably smaller adsorption energy with the substrate, result in non-deformed particle after adsorption, as it happens with other virus capsids (*Carrasco et al., 2009*).

In this mechanical study of a multi-layered virus we have shown how the biophysical properties and interactions of the three particle shells are finely tuned to produce an infective RV virion. While the high mechanical strength provided by the strong VP7-VP7 and VP7-VP6 interactions (*Figure 7H*, black springs) relates to protection tasks, the lower resistance of the VP6-VP6 and VP6-VP2 interactions (*Figure 7H*, white springs) guarantees the conformational dynamics required for transcription. Importantly, the interference with this finely tuned mechanical regulation offers new venues for development of antiviral strategies.

# Materials and methods

### Key resources table

| Reagent type (species)or resource | Designation | Source or reference | Identifiers | Additional information |
| --- | --- | --- | --- | --- |
| Cell line (Chlorocebus aethiops) | MA104 | ECACC Cat# 85102918 | RRID:CVCL_3845 | |
| Biological sample (Rotavirus A) | SA11-C4111 | PMID: 11913378 | GenBank:KJ450831; KJ450832;KJ450833; KJ450834;KJ450835; KJ450836;KJ450837; KJ450838;KJ450839; KJ450840;KJ450841 | |
| Software, algorithm | Xmipp | PMID: 15477099 | | http://xmipp.cnb.csic.es /twiki/bin/view/Xmipp/WebHome |
| Software, algorithm | RELION | PMID: 23000701 | RRID:SCR_016274 | https://www2.mrc-lmb.cam.ac.uk/ relion/index.php?title=Main_Page |
| Software, algorithm | COMSOL Multiphysics 4.3 | Comsol, Stockholm, Sweden | RRID:SCR_014767 | |
| Software, algorithm | CTFFIND3 | PMID: 12781660 | | http://grigorrieflab.janelia.org/ctf |
| Software, algorithm | Delphi | PMID: 11913378 | | http://honig.c2b2.columbia.edu/delphi/ |
| Software, algorithm | UCSF Chimera | PMID: 24873828 | RRID:SCR_004097 | https://www.cgl.ucsf.edu/chimera/ |

## TLP, DLP and SLP production and purification

The simian rotavirus strain SA11-C4111 (*Rodríguez et al., 2014*) was used in this study. Viruses were grown using the monkey epithelial cell line MA104 (ECACC 85102918), cultured in MEM with 10% fetal calf serum, and used between passages 10 and 25. The amplified viruses were used within three passages of the last plaque isolation step.

For the production of TLP, 3 day post-confluent monolayers of MA104 cells were infected with a multiplicity of 0.5 PFU/cell. Activation of the viruses was performed for 30 min at 37°C with 100 BAEE U/ml of TPCK-treated trypsin (TPCK Trypsin, Thermo Scientific Pierce). To remove serum, cell monolayers were washed twice with MEM prior to absorption (60', 37°C). After absorption, monolayers were washed with MEM and incubated in MEM containing 10 BAEE U/ml TPCK-trypsin. Cells and extracellular media were harvested when total cytopathic effect was observed. TLP were purified from these extracts as previously indicated (*Rodríguez et al., 2014*). Purified TLP were diluted to 0.2 mg/ml of protein content in 1xTNC (10 mM Tris:HCl pH 7.5, 140 mM NaCl, 10 mM $CaCl_2$) containing 10% glycerol and 0.02% sodium azide, flash frozen in liquid nitrogen as small (5 μl) aliquots, and stored at −80°C.

The preparation of DLP from purified TLP by treatment with EDTA at 37°C and its isolation in CsCl gradients has been performed as described by Patton et al (*Patton et al., 2000*). Purified DLP were diluted to 0.2 mg/ml of protein content in 1xTNE (10 mM Tris:HCl pH 7.5, 140 mM NaCl, 1 mM EDTA) containing 10% glycerol and 0.02% sodium azide flash frozen in liquid nitrogen as small (5 μl) aliquots, and stored at −80°C.

SLP were prepared form purified DLP by treatment with 1.25M $CaCl_2$ in a solution containing 0.75M trehalose, 0.15M NaCl, 20 mM Borate buffer (pH 8.45) and Complete-EDTA Free protease inhibitors (Roche) at the manufacturer recommended concentration. DLP, at a concentration of 100 μg/ml, where incubated for 2 hr at 37°C with gentle agitation. After the treatment, the concentration of trehalose in the mixture was reduced to 0.25M by dilution with two volumes of the buffer without trehalose, and incubated at room temperature (22°C) during 90 min, with gentle agitation. SLP were concentrate by centrifugation (20.000 g, 60 min, 22°C) and resuspended in a buffer containing 1.50M trehalose, 0.15M NaCl, 20 mM Tris:HCl (pH 8.45) and Complete-EDTA Free protease inhibitors (Roche) at the manufacturer recommended concentration. Purified SLP were diluted to 0.2 mg/ml in 1xTNC containing 0.5M trehalose, flash frozen in liquid nitrogen as small (5 μl) aliquots, and stored at −80°C.

## Electron microscopy and image processing

For transmission electron microscopy, purified particles were applied to glow-discharged carbon-coated grids and negatively stained with 2% aqueous uranyl acetate. Images were recorded on a Gatan 1 k CCD camera in a FEI Tecnai 12 microscope operated at 120 kV.

For cryo-EM, samples were applied to Quantifoil R 2/2 holey grids, blotted, and plunged into liquid ethane using a Leica EM CPC cryo-fixation unit. Cryo-EM images were recorded in low-dose conditions (~10 e$^-$/Å [*Müller et al., 2002*]) on a FEI Eagle 4 k CCD using a Tecnai G2 electron microscope operating at 200 kV and a detector magnification of 67,873X (2.16 Å/pixel sampling rate).

Image processing operations were performed using Xmipp (*Marabini et al., 1996*) and Relion (*Scheres, 2012*) and graphic representations were produced by UCSF Chimera (*Pettersen et al., 2004*). Xmipp automatic picking routine was used to select 4238 particles and defocus was determined with CTFfind3 (*Mindell and Grigorieff, 2003*). Images were 2D classified using the corresponding Relion routine to select 4200 homogenous particles. To avoid any bias at the spike density, the published structure of the rotavirus VP7-recoated particle (*Chen et al., 2009*), low-pass filtered to 30 Å, was used as initial model for Relion to obtain a 3DR using the corresponding Relion autorefinement routine. Resolution was assessed by gold standard Fourier Shell Correlation (FSC) between two independently processed half datasets. Applying a correlation limit of 0.5 (0.3), the resolution is 14.2 (12.6).

The electrostatic potentials were calculated using DelPhi software (*Rocchia et al., 2002*) and surface-colored with UCFS Chimera.

## AFM experiments

Measurements were carried out with an AFM (Nanotec Electrónica S.L., Madrid, Spain) operating in Jumping Mode Plus (*Ortega-Esteban et al., 2012*). This intermittent-contact imaging mode consists on performing low force-versus-Z-piezo-displacement (FZ) curves at every point of the imaging area, with nanometric lateral movements of the sample where it is far (~40 nm) from the tip. All the experiments were carried out with rectangular silicon-nitride cantilevers (RC800PSA, Olympus, Tokyo, Japan) with nominal springs constants of 0.05 N/m, and were routinely calibrated using the Sader's method (*Sader et al., 1999*). The obtained images were processed using the WSxM software (*Horcas et al., 2007*).

For adsorption of particles, one 5 μl aliquot of particles was thawed on ice and diluted to 50 μl with TNC (for TLP and DLP) or in TNC-Trehalose (for SLP). They were incubated for 15 min on freshly cleaved highly oriented pyrolytic graphite (HOPG; ZYA quality; NT-MDT, Tempe, AZ). The non-adsorbed particles were removed by performing several washes consisting in the addition of 50 μl of TNC and the extraction of 50 μl of the sample. The tip was also prewetted with a 20 μl drop of TNC before starting the image acquisition process.

For single nanoindentation assays, individual particles were deformed with the AFM tip by performing single force curves at a constant speed (150 nm/s) and with a high Z piezo displacement (150 nm) to ensure that the tip always reached the substrate after the disruption of the particle. Images before and after the FZ were obtained to observe the structural damages suffered by each particle. The mechanical properties (elastic constant, breaking force and critical strain) were obtained from these FZ curves.

For cyclic loading assays, the topographic image acquisition with the AFM tip was used to mechanically fatigue single particles (TLP, DLP and SLP), causing their guided disassembly and allowing to image the dynamics of the process. The number of scanning points in the 'x' and 'y' coordinates (128 in each direction), and the size of the image (~300 nm) were established to apply one loading cycle each ~2–3 nm.

Real time experiments of TLP disassembly while removing Ca ions was carried out as follows.

TLP were initially in the AFM liquid chamber with 70 μl of TNC buffer. This chamber was connected to two syringe pumps (NE-1000, New Era Pump Systems, Inc.). One of the syringes was used for pumping TNE buffer into the chamber while, simultaneously, the other syringe was withdrawing liquid. The pumping/withdrawing rate was 1 μl/min, and the fatigue experiments lasted ~80 min. Under these conditions TNC buffer was totally replaced by TNE at the end of the experiment, thus ensuring the chelation of all the Ca2 +ions initially present in the TLP.

## Finite Element (FE) simulations of rotavirus

Finite elements simulations mimicking the AFM nanoindentation of the different rotavirus particles were performed using the program COMSOL Multiphysics 4.3 (Comsol, Stockholm, Sweden). In the simulations, each layer was modeled as a homogenous spherical shell made of a material with Young's modulus E and Poisson ratio $\nu = 0.3$ (a standard value for protein-like materials). This model shell was placed on a hard flat substrate and indented by a hard spherical object with radius $R_{in} = 15$ nm, mimicking the nominal radius of the AFM tip. The system was simulated using a 2D axisymmetric model that was meshed with over 1400–6000 triangular elements. The contacts between the shell and the tip as well as the supporting surface during indentation were implemented with a contact normal penalty factor. This parameter controls the hardness of the interface surface and it is used to prevent the penetration of the two boundaries coming into contact. The penalty factor used was $Y/\Delta x$, where Y is the Young's modulus and $\Delta x$ is the minimum element size of the mesh of the material which is indented. A parametric, non-linear solver was used to simulate the stepwise lowering of the tip onto the capsid. The spring constant was obtained in all cases from the slope of the force versus indentation curves at a small value of the indentation of 2 nm. For multilayer shells, two different cases were simulated: a model in which the shells are joint and coupled (using the COMSOL option Union to finalize the geometry), and a second case in which the layers are independent and uncoupled (using the option Assembly to finalize the geometry). In both cases, the results for the stress distribution, the force-indentation curves and the spring constant for small indentations were identical.

The error bars in the values of Young's modulus for the different layers were calculated in the FEM simulations in the following way. For each value of the experimental spring constant $k \pm \delta k$, we did FEM simulations to find which value of the Young's modulus, Y, was giving a slope of k; which value, $Y_{min}$, was giving a slope $k-\delta k$; and which value, $Y_{max}$, was yielding $k+\delta k$. The best estimate and approximated uncertainty in the Young's modulus were reported as $Y \pm (Y_{max}-Y_{min})/2$. The SLP was modeled as a spherical shell with an external radius R = 27 nm and thickness h = 3.5 nm (see inset in *Figure 4—figure supplement 1*). A Young's modulus of $Y_1 = 0.53 \pm 0.20$ GPa was used in order to recover the same slope in the simulations as the one measured experimentally.

The DLP was modeled as a double-layer spherical shell with an external radius R = 35 nm, made of an outer layer with Young's modulus $Y_2 = 0.0815 \pm 0.070$ GPa and thickness h = 8.0 nm, and an inner layer with Young's modulus $Y_1 = 0.53 \pm 0.20$ GPa and thickness h = 3.5 nm (see inset in *Figure 4—figure supplement 1*).

Finally, the TLP was modeled as a triple-layer spherical shell, by adding a third layer with Young's modulus $Y_3 = 1.0 \pm 0.9$ GPa and thickness h = 3.5 nm, mimicking the VP7 (see inset in *Figure 4—figure supplement 1*).

## Acknowledgements

This work was supported by grants from the Spanish Ministry of Economy and Competitivity FIS2014-59562-R, FIS2017-89549-R and 'María de Maeztu' Program for Units of Excellence in R and D (MDM-2014–0377) to PJP, BFU2013-43149-R to DL and JMR, FIS2015- 67837 P to DR, FIS2015-71108-REDT to PJP, DL, DR, and JMR, and BFU2014-55475-R (Spanish Ministry of Economy and Competitivity) and S2013/MIT-2807 (Comunidad Autónoma de Madrid) to JRC.

## Additional information

### Funding

| Funder | Grant reference number | Author |
| --- | --- | --- |
| Ministerio de Economía y Competitividad | FIS2014-59562-R | Manuel Jiménez-Zaragoza<br>Marina PL Yubero<br>Pedro J de Pablo |

| Ministerio de Economía y Competitividad | FIS2015-71108-REDT | Manuel Jiménez-Zaragoza<br>Marina PL Yubero<br>Jose R Castón<br>David Reguera<br>Daniel Luque<br>Pedro J de Pablo<br>Javier M Rodríguez |
| --- | --- | --- |
| Ministerio de Economía y Competitividad | FIS2017-89549-R | Manuel Jiménez-Zaragoza<br>Marina PL Yubero<br>Pedro J de Pablo |
| Ministerio de Economía y Competitividad | BFU2014-55475-R | Jose R Castón |
| Comunidad Autónoma de Madrid | S2013/MIT-2807 | Jose R Castón |
| Ministerio de Economía y Competitividad | FIS2015- 67837 P | David Reguera |
| Ministerio de Economía y Competitividad | BFU2013-43149-R | Daniel Luque<br>Javier M Rodríguez |
| Ministerio de Economía y Competitividad | MDM-2014–0377 | Pedro J de Pablo |

The funders had no role in study design, data collection and interpretation, or the decision to submit the work for publication.

## Author contributions

Manuel Jiménez-Zaragoza, Data curation, Formal analysis, Investigation, Methodology, Writing—original draft; Marina PL Yubero, Formal analysis, Investigation, Methodology; Esther Martín-Forero, Investigation, Methodology, Writing—review and editing; Jose R Castón, Conceptualization, Resources, Formal analysis, Funding acquisition, Methodology, Writing—review and editing; David Reguera, Conceptualization, Resources, Software, Investigation, Methodology, Writing—original draft; Daniel Luque, Conceptualization, Resources, Data curation, Formal analysis, Supervision, Funding acquisition, Validation, Investigation, Visualization, Methodology, Writing—original draft, Writing—review and editing; Pedro J de Pablo, Conceptualization, Resources, Data curation, Formal analysis, Supervision, Funding acquisition, Validation, Investigation, Visualization, Methodology, Writing—original draft, Project administration, Writing—review and editing; Javier M Rodríguez, Conceptualization, Data curation, Formal analysis, Supervision, Funding acquisition, Validation, Investigation, Methodology, Writing—original draft

## Author ORCIDs

Manuel Jiménez-Zaragoza http://orcid.org/0000-0003-4739-699X
Marina PL Yubero https://orcid.org/0000-0003-3751-4702
David Reguera http://orcid.org/0000-0001-6395-6112
Daniel Luque http://orcid.org/0000-0002-0151-6020
Pedro J de Pablo http://orcid.org/0000-0003-2386-3186
Javier M Rodríguez https://orcid.org/0000-0003-0146-9903

## Decision letter and Author response

Decision letter https://doi.org/10.7554/eLife.37295.035
Author response https://doi.org/10.7554/eLife.37295.036

# Additional files

## Supplementary files

• Transparent reporting form
DOI: https://doi.org/10.7554/eLife.37295.033

## Data availability

Excel documents have been provided with the data of the graphs for Figures 2-6, as well as Figure 4-figure supplement 1 and Figure 5-figure supplement 1.

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
