## [Decision Letter]

Thank you for submitting your article "Mechanics of rotavirus particles accounts for the functions of protein shells in a multilayered virus" for consideration by *eLife*. Your article has been reviewed by three peer reviewers and the evaluation has been overseen by a Reviewing Editor and Arup Chakraborty as the Senior Editor. The following individual involved in review of your submission has agreed to reveal his identity: Ulrich Desselberger (Reviewer #3).

The reviewers have discussed the reviews with one another and the Reviewing Editor has drafted this decision to help you prepare a revised submission.

Summary:

The manuscript "Mechanics of rotavirus particles accounts for the functions of protein shells in a multilayered virus" describes nano indentation experiments performed on a multi-shell dsRNA virus complemented by finite-element modeling. The three reviewers are unanimous in considering this work to be a significant advance for understanding the rotavirus (RV) life cycle in the context of the mechanical properties of the RV protein layers. The rotavirus system is very different from other viruses that have been studied by AFM indentation. The potential interplay between the three protein layers that enclose the genome is fascinating, and the manuscript makes for a very interesting read. Mechanical probing is performed systematically on 3-layer, 2-layer, and single layer particles, resulting in a comprehensive set of characteristics (stiffness, breaking force, fatigue). The evolution of mechanical fatigue pattern of TLPs during treatment with EDTA is well-studied. The functions proposed for the role of Ca^2+^ in the TLPs are of interest. The work is of fundamental importance for RVAs, probably also for other genera of the Reoviridae, and possibly beyond for other segmented RNA virus families.

There are, however, some points that need to be addressed in order to be considered further for publication in *eLife*.

Essential revisions:

1) The distribution of particle heights about the mean is broad (~15 nm). The total spike extension observable in Figure 2D is 5 nm at most. Yet, the authors explain the wide height distribution as an effect of spike orientation or presence. Moreover, even when there are no spikes, the height distribution seems to be considerable for DLPs, but not so for SLPs. There seems to be something additional involved here than the spikes. Can the authors please explain?

2) The fatigue experiment is interesting. It probes the activation energy barrier for deformation in a way that temperature could not. However, it is not clear why a large breaking force is expected to correlate with a high activation energy barrier as the authors claim. Perhaps the authors could elaborate on this point.

3) Can the authors please elaborate on the FEM results and analysis as follows:

– The indentation model appears to be implemented as three independent layers. However, evidence is presented for Ca-mediated inter-layer interaction, or coupling, and data interpretation (Figure 6H) relies on it. There is no discussion of shear/coupling effects in the manuscript, which play a well known role in the rigidity of bilayer membranes. Specifically, the energy required to deform coupled shells is expected to be greater (2-4 times) than that for uncoupled shells. This is due to shear not propagating radially between uncoupled shells. The manuscript would improve qualitatively if the apparent discrepancy between an uncoupled layer model and the coupled layer evidence is addressed.

– Can the authors please clarify the last line of the caption to Figure 4—figure supplement 2, in the caption and/or through additional text in the Materials and methods description (subsection “Finite Element (FE) simulations of rotavirus”)? How were the error bars in the Young's modulus values calculated? In Figure 4—figure supplement 2, can you compare the best fit FEM results to the experimental average for each particle type? How sensitive are the results to parameters such as the height of each layer? This could be relevant, since the actual shells are spherically inhomogeneous in height.

– Can the authors justify the assumption that the interior contents (genome and transcriptional machinery) have no effect on the particle's response to deformation. In some previous experiments internal contents have been shown to affect mechanical properties of viral particles; e.g. double stranded DNA inside of bacteriophage capsids, or different RNA molecules in Brome mosaic virus particles. While the effects of double-stranded DNA could be attributed to the internal pressure, the effects of different BMV RNA molecules must involve more subtle interactions.

4) The authors' finding of the key role of Ca^2+^ ions in mediating several protein-protein interactions is interesting. Is it possible to explain why Ca chemistry enables these interactions, and to compare these Ca-mediated interactions to the roles of Ca in other virus systems?

---

## [Author Response]

Essential revisions:1) The distribution of particle heights about the mean is broad (~15 nm). The total spike extension observable in Figure 2D is 5 nm at most. Yet, the authors explain the wide height distribution as an effect of spike orientation or presence. Moreover, even when there are no spikes, the height distribution seems to be considerable for DLPs, but not so for SLPs. There seems to be something additional involved here than the spikes. Can the authors please explain?

The height data of TLP are referred to the highest point of the topographic profile, excluding the spikes located at the top of the particle. In particular, the TLP profile of Figure 2D (red) results in a height of 78 and 83 nm excluding and including the spikes height, respectively. In this case the selected data point is 78 nm (dashed line). We did so because spikes at the top are unstable and easily modified by the tip. Many times they are not even present. We have included this explanation in the caption of Figure 2. With this consideration, the height distribution for TLP height (69.7 ± 6.1 nm) is not that broad since the standard deviation is below 9% of the height value. Interestingly, the height data points of TLP (Figure 2E, red) suggest two populations centered at ~74 and ~62 nm, represented by filled and empty symbols, respectively. We propose that these data correspond with the presence (red filled squares Figure 2E) or absence (red empty squares, Figure 2E) of spikes at the particle-surface interface. In the first case the presence of spikes would prevent partially the contact between the VP7 layer and the substrate (Figure 2—figure supplement 2), thus precluding virus adsorption and deformation. In the second case, we hypothesize that the thin VP7 layer interacts strongly with HOPG to induce the deformation of the soft and thick VP6 subjacent layer. We revise this topic in the new Figure 2—figure supplement 2.

Again, the DLP height distribution (65.7 ± 2.8 nm) is not that broad if we consider that the standard deviation is below 5% of the height value, which is similar to other viruses (Llauro et al., 2016). VP6 layer constitutes the thickest and softest layer of RV particle, implying a high deformability upon adsorption. Strikingly, DLP average height value is slightly bigger than the lowest population of TLP that we ascribed to particles without spikes at the virus-surface interface. We propose that the VP7-HOPG interaction is so strong that deforms both the VP7 and VP6 layers of TLP. SLP height value (53.8 ± 0.9 nm) shows a standard deviation below 2% of the height value. The low collapse exhibited by SLP is probably due to its high effective Young’s modulus. In summary, besides the presence or absence of spikes, the spread in heights is probably due to the different levels of adsorption of virus particles to the substrate, which determines the degree of collapse as explained in Zeng et al., (2017).

We have included text in the first paragraph of the subsection “AFM topography of TLP and RV subviral particles”, and in the third paragraph of the Discussion and new Figure 2—figure supplement 2 discussing these topics.

2) The fatigue experiment is interesting. It probes the activation energy barrier for deformation in a way that temperature could not. However, it is not clear why a large breaking force is expected to correlate with a high activation energy barrier as the authors claim. Perhaps the authors could elaborate on this point.

With breaking force we refer to how much force should be applied to break the virus shell in a single nanoindentation. Since the tip and virus radius are very similar, the breaking force is probing the bonds between most proteins building the virus. In contrast, the indentation used in fatigue is ~10 times less, deforming individual capsomers in a local fashion. A strong binding energy between capsomers would not only result in a high resistance of individual proteins against fatigue, but also will contribute to a high breaking force when all capsomers are probed in a single indentation assay experiment. It is like a building with the bricks well cemented between them will show a high resistance to be demolished by a wrecking ball. We have found similar results in lambda phage (Hernando-Pérez, et al., 2014). We have included text in the first paragraph of the subsection “Mechanical fatigue” explaining these ideas.

3) Can the authors please elaborate on the FEM results and analysis as follows:– The indentation model appears to be implemented as three independent layers. However, evidence is presented for Ca-mediated inter-layer interaction, or coupling, and data interpretation (Figure 6H) relies on it. There is no discussion of shear/coupling effects in the manuscript, which play a well known role in the rigidity of bilayer membranes. Specifically, the energy required to deform coupled shells is expected to be greater (2-4 times) than that for uncoupled shells. This is due to shear not propagating radially between uncoupled shells. The manuscript would improve qualitatively if the apparent discrepancy between an uncoupled layer model and the coupled layer evidence is addressed.

This is a very interesting point. We have simulated using FEM both the cases in which the different layers are coupled and uncoupled. More specifically, this was accomplished in COMSOL by finalizing the geometry using Union, which will join and couple the interface between the different layers, and ASSEMBLY, which will treat the different layers as independent and uncoupled. In both cases, the resulting indentations, stress distributions and effective spring constants were identical. We also added a small gap between layers and an additional contact penalty to simulate their uncoupling, finding the same results.

Protein shells are different from lipid bilayers. Unlike lipid bilayers, protein shells are not fluid and have effective Young’s modulus significantly higher (on the order of GPa instead of MPa). In addition, nanoindentation experiments mostly involve normal deformations. The difference between coupled an uncoupled viral protein shells will be more evident when the virus is subjected to shear deformations (e.g. by applying lateral shear forces with the tip). In such a case, not only the coupling but the adhesion strength between the different layers will be very relevant. But apparently, for small polar indentations of solid-like shells, there seems to be no difference between coupled and uncoupled shells. We have added a discussion of that interesting point in the Materials and methods section of the revised manuscript (subsection “Finite Element (FE) simulations of rotavirus”, first paragraph).

– Can the authors please clarify the last line of the caption to Figure 4—figure supplement 2, in the caption and/or through additional text in the Materials and methods description (subsection “Finite Element (FE) simulations of rotavirus”)? How were the error bars in the Young's modulus values calculated? In Figure 4—figure supplement 2, can you compare the best fit FEM results to the experimental average for each particle type? How sensitive are the results to parameters such as the height of each layer? This could be relevant, since the actual shells are spherically inhomogeneous in height.

We have modified the last line in the caption of Figure 4—figure supplement 2 to clarify that the value of the Young’s modulus for each layer was chosen in such a way that the slope of the resulting indentation curve will be the same as the effective spring constant found in the experiments. The details are provided in the Materials and methods section.

The error bars in the values of Young’s modulus for the different layers were calculated in the FEM simulations in the following way. For each value of the experimental spring constant k ± δk, we did FEM simulations to find which value of the Young’s modulus, Y, was giving a slope of k; which value, Y_min_, was giving a slope k-δk; and which value, Y_max_, was yielding k+δk. The best estimate and approximated uncertainty in the Young’s modulus was reported as Y ± (Y_max_-Y_min_)/2. We have included this explanation in the Materials and methods subsection “Finite Element (FE) simulations of rotavirus”, first paragraph.

Following the suggestion of the reviewers, we have modified Figure 4—figure supplement 2 to include experimental data of typical nanoindentation curves for TLP, SLP and DLP.

The value of the effective Young’s modulus is certainly sensitive to the height of each layer and the actual viral shells are indeed inhomogeneous in height. The strongest influence of inhomogeneities in the shell is through the thickness. For a given effective spring constant, the effective Young’s modulus inferred from it goes as 1/h^2^. Accordingly, uncertainties or variations in the height of magnitude δh will lead to relative uncertainties in the Young’s modulus of δY/Y= 2 δh/h. The relatively high variance in the experimental results translates in high uncertainties in Young’s modulus. The relative uncertainties reported in Young’s modulus (38%, 86% and 90% for VP2, VP6 and VP7, respectively) will account for variations in height of 19%, 43%, and 45%, which are indeed much larger than the actual height inhomogeneities in the real shells. Therefore, the uncertainty in the evaluation of the Young’s modulus due to shell inhomogeneities is already accounted for in the error bars coming from the experiments (subsection “Finite Element (FE) simulations of rotavirus”, first paragraph).

– Can the authors justify the assumption that the interior contents (genome and transcriptional machinery) have no effect on the particle's response to deformation. In some previous experiments internal contents have been shown to affect mechanical properties of viral particles; e.g. double stranded DNA inside of bacteriophage capsids, or different RNA molecules in Brome mosaic virus particles. While the effects of double-stranded DNA could be attributed to the internal pressure, the effects of different BMV RNA molecules must involve more subtle interactions.

Unfortunately, we cannot experimentally have access to empty SLPs. Therefore, we cannot directly compare the mechanical response of empty *versus* full SLPs, to assess the importance of the interior contents on the particle’s response to deformation. However, we can provide an estimate based on theoretical arguments and comparison with other viruses. Eleven segments of dsRNA, with a total genome size of 18.5 kbp, are the major component of the core inside the VP2 layer. Using the approximations of Purohit et al. (2005) to estimate the packing fraction, gives a value of ~24%, which is below the typical values for many pressurized tailed bacteriophage viruses, such as phi29 (46%), T7(49%), T4(44%) or lambda (42%). Accordingly, we expect that the influence of the core on the particle mechanics would be relatively low, compared to these other viruses. A simple estimate of the internal pressure using the inverse spool model yields a value below 10 atm. In fact, a pressurized genome inside the SLP of 10 atm would increase the spring constant from empty to full SLP in about 10% (see Hernando-Perez et al., 2012), that is well below the error bars of our measure of k_SLP_ and of our estimations for the Young’s Modulus.

In any case, the presence of the core would only affect to the estimation of the Young’s Modulus of the VP2 layer, but not to the inferred properties of the VP6 and VP7 layers.

We have added text to the subsection “Finite elements analysis of TLP and RV subviral particles” commenting on this.

4) The authors' finding of the key role of Ca^2+^ ions in mediating several protein-protein interactions is interesting. Is it possible to explain why Ca chemistry enables these interactions, and to compare these Ca-mediated interactions to the roles of Ca in other virus systems?

Following the reviewer suggestion, we have added text to the manuscript explaining the role of Ca^2+^ ions in the stabilization of VP7 trimers (Discussion, second paragraph). We have also included references to the unique coordination chemistry of Ca that allows for these interactions, and examples of the role of Ca in other viral systems.